# Silver Nanoparticle-Intercalated Cotton Fiber for Catalytic Degradation of Aqueous Organic Dyes for Water Pollution Mitigation

**DOI:** 10.3390/nano12101621

**Published:** 2022-05-10

**Authors:** Matthew Blake Hillyer, Jacobs H. Jordan, Sunghyun Nam, Michael W. Easson, Brian D. Condon

**Affiliations:** Cotton Chemistry and Utilization Research Unit, Southern Regional Research Center, Agricultural Research Service, United States Department of Agriculture, New Orleans, LA 70113, USA; matthew.hillyer@usda.gov (M.B.H.); jacobs.jordan@usda.gov (J.H.J.); michael.easson@usda.gov (M.W.E.); brian.condon@usda.gov (B.D.C.)

**Keywords:** catalysis, silver nanoparticles, environmental, health issues, pollution remediation, cotton fibers, water

## Abstract

Azo dyes are commonly used in textile color processing for their wide array of vibrant colors. However, in recent years these dyes have become of concern in wastewater management given their toxicity to humans and the environment. In the present work, researchers remediated water contaminated with azo dyes using silver nanoparticles (Ag NPs) intercalated within cotton fabric as a catalyst, for their enhanced durability and reusability, in a reductive degradation method. Three azo dyes—methyl orange (MO), Congo red (CR), and Chicago Sky Blue 6B (CSBB)—were investigated. The azo degradation was monitored by UV/vis spectroscopy, degradation capacity, and turnover frequency (TOF). The Ag NP–cotton catalyst exhibited excellent degradation capacity for the dyes, i.e., MO (96.4% in 30 min), CR (96.5% in 18.5 min), and CSBB (99.8% in 21 min), with TOFs of 0.046 min^−1^, 0.082 min^−1^, and 0.056 min^−1^, respectively, using a 400 mg loading of catalyst for 100 mL of 25 mg L^−1^ dye. To keep their high reusability while maintaining high catalytic efficiency of >95% degradation after 10 cycles, Ag NPs immobilized within cotton fabric have promising potential as eco-friendly bio-embedded catalysts.

## 1. Introduction

Increasing consumer demand for affordable apparel has put significant economic stress on textile processing, resulting in the dramatic rise of pollutants in wastewater effluent [1,2,3]. One of the major contaminants found in clothing manufacturing water discharge is organic dyes. All organic dye chemical structures are based on at least 1, up to a combination of 25, types of functional groups resulting in tens of thousands of possible chemical structures [4]. Of these many structures, the most common type used by material consumed in processing is azo-containing dyes. Azo dyes consist of at least one diazenyl (–N=N–) functional group. Total azo-dye use constitutes up to 70%, or 9 million tons, of total annual dye consumption worldwide [5,6]. Of this, 4.5 million tons of dye and dye degraded byproducts are disposed of annually, posing severe environmental and anthropogenic hazards.

There have been numerous studies showing the ecotoxicity of dyes in aquatic and terrestrial environments [7,8,9,10,11,12]. Water containing high concentrations of azo-dye decrease sun permeability, resulting in diminished photosynthesis by marine plants, significantly reducing the amount of available oxygen and causing ecosystem collapse [13]. Research focusing on specific micro- and macro-organisms found acute toxicity leading to chronic effects, including those caused by bioaccumulation, mutagenicity, and carcinogenicity [14,15,16]. Azo dyes can also be deleterious to human health via two modalities of exposure: skin contact and ingestion [17]. Skin contact by azo dyes, specifically through colored garments, can cause allergic reactions, dermatitis, and skin irritation [18]. Ingestion of azo dyes causes damage to the endocrine and gastrointestinal systems and has been directly linked to several cancers, including those originating in the kidneys, liver, and bladder [17,19,20]. Therefore, there is great demand for an efficient and effective method to degrade azo dyes in industrial waste streams and municipal water sources.

Conventional industrial processes for the treatment of azo dye-contaminated wastewater include isolation (flocculation, coagulation, or membrane) or degradation (chemical or microbial) [21]. Isolation is often operationally costly and requires non-reusable coagulants or flocculants to form a physicochemical sludge [22]. This byproduct normally involves additional processing, further adding to the cost-prohibiting aspect. Recently, attention has been paid to using genetically engineered microorganisms as an avenue to remediate dye in industrial wastewater [23]. While this method generates a quantity of sludge, conditions must be kept optimal for microbe viability. Additionally, the process is often slow and the selectivity of certain molecules is often low. Alternatively, chemical degradation can be highly efficient and, with the application of catalysts, the reactions can proceed under mild conditions at ambient temperatures. There have been several studies investigating the highly effective degradation of azo dyes using silver nanoparticles (Ag NPs) [24,25,26,27]. While these methods are effective, they rely on using Ag NPs in free solution. Furthermore, the isolation and extraction of the catalyst presents a problem for cost-effective reusability. Herein, we present studies using a durable cotton-embedded Ag NP catalyst for the degradation of three azo dyes: methyl orange (MO), Congo red (CR), and Chicago Sky Blue 6B (CSBB) (Table 1). The capacity of the catalyst to be used over several iterations of the reaction cycle was investigated using simple extraction and isolation.

## 2. Materials and Methods

### 2.1. Chemicals

Silver nitrate (99.9999%, trace metal basis), ammonium hydroxide (40% *w*/*v*), sodium hydroxide (50% *w*/*v*), sodium borohydride (≥98%), Congo red (dye content ≥85%), methyl orange (dye content 85%), Chicago Sky Blue B (dye content ~80%), methyl methacrylate, butyl methacrylate, and methyl ethyl ketone were purchased from Sigma-Aldrich (St. Louis, MO, USA). All chemicals were used without further purification. Bleached and desized print cloth was purchased from Testfabrics, Inc. (West Pittston, PA, USA). Deionized water was used in the synthesis of Ag NPs in cotton textiles and in the preparation of dye solutions. All glassware was cleaned with 28% nitric acid and rinsed several times with deionized water.

### 2.2. Preparation and Characterization of Ag NP–Cotton Catalyst

The synthetic procedure for producing Ag NP-intercalated cotton fabric was used from a previously reported method [28]. A 5 cm × 10 cm specimen was immersed into 20 mL of an aqueous NaOH solution (20 *w*/*v*) and agitated at room temperature for 15 min. The excess NaOH solution was removed using a laboratory padder (Werner Mathis USA Inc., Concord, NC, USA) at a pressure of 0.3 MPa and a speed of 2 m/min. The resulting wet pick-up was 185 ± 5%. The fabric was transferred to 30 mL of an aqueous solution containing silver nitrate (14.7 mM) and ammonium hydroxide solution (60.5 mM) and agitated at room temperature for 15 min. The excess silver precursor solution was removed from the fabrics using the same padding method. The fabric was then transferred into 50 mL of an aqueous L-ascorbic acid solution, in which the fabric immediately turned brown in color. The obtained fabric was washed in DI water multiple times and air dried.

X-ray diffraction (XRD) spectra of pristine and Ag NP cotton fibers were collected using a PANalytical Empyrean X-ray Diffractometer (Malvern Panalytical, Malvern, UK) with CuKα radiation (1.54060 Å) and 45 kV and 40 mA for generator settings. Angular scanning was collected from 8.0 to 60° 2θ with a step size of 0.03° and rate of 0.6°/min. X-ray photoelectron spectroscopy (XPS) measurements were collected using a VG Scientific ESCALAB MKII (Thermo Scientific, Waltham, MA, USA) spectrometer system using a Mono AlKα X-ray excitation source (*hν* = 1486.6 eV) at 450 W power. The chamber pressure was below 3 × 10^−9^ mbar during analysis. Data acquisition was collected with 6 s dwell times per data point. The dwell time per data point was increased to 60 s per point for C1s and O1s, and 200 s per point for Ag3d to improve the signal-to-noise ratio at regions of interest. The signal of each data point was rescaled to dwell times of 60 s after data acquisition. The spectrum was calibrated by reference to the C 1s peak at 284.8 eV binding energy. All data was processed with Casa XPS software (Version 2.3.24, Casa Software, Ltd., Teignmouth, UK).

The uniform internal dispersion of Ag NPs within the cotton fiber was confirmed by a transmission electron microscope (TEM) (FEI Technai G2 F30) operating at 300 kV using a procedure developed at the Southern Regional Research Center located in New Orleans, LA, USA [29,30]. Fibers from the functionalized Ag NP-embedded cotton fabric were combed prior to immersion in a methacrylate matrix solution in Teflon tubing (3.2 mm inner diameter), and then polymerized under UV light for 30 min. The resulting block of coated fibers was removed from the tubing and subsequently immobilized in polyethylene capsules. The fibers were sectioned into 100–120 nm slices using a PowerTome Ultramicrotome (Boeckeler Instruments, Inc., Tucson, AZ, USA). The cross-sectioned slices were affixed to a copper grid coated with carbon film, and the polymer block was dissolved using methyl ethyl ketone.

The concentration of silver in the Ag NP cotton fabric was determined using inductively coupled plasma mass spectrometry (ICP-MS) at the University of Utah ICP-MS Metals Lab. UV/vis absorbance spectra were obtained of the Ag NP cotton for the wavelength range of 300 to 750 nm with a step size of 1.0 nm using a UV/vis/NIR spectrophotometer (ISR-2600, Shimadzu, Columbia, MA, USA). Origin 2018b Graphing and Analysis Software (OriginLab, Northampton, MA, USA) was used for analysis of spectral data.

### 2.3. Catalytic Degradation Studies and Turnover Determinations

Calibration curves were obtained for each dye within the range from 0.1 mg L^−1^ to 50 mg L^−1^ by UV/vis spectroscopy; for which 0.1 mg L^−1^ is the limit of detection for each dye, and beyond 50 mg L^−1^ the absorbance is overexposed and deviates from linearity. The UV/vis spectra were collected for a wavelength range of 300–750 nm and a step size of 1.0 nm using a UV/vis/NIR spectrometer (ISR-2600, Shimadzu, Columbia, MA, USA). Acrylic cuvettes with a 1 cm optical path length filled with 3.0 mL solution were used.

For general catalytic degradation studies, 30.0 mg sodium borohydride (NaBH_4_, 0.777 mmol) was dissolved in a 100 mL solution of 25 ppm dye (3.59 µmol CR, 7.64 µmol MO, 2.52 µmol CSBB) and stirred using a magnetic stirrer bar at 180 rpm at room temperature. To this solution, either 200 mg or 400 mg Ag NPs–cotton catalyst (13,150 ppm Ag) was added, and the reaction time was started. The UV/vis spectrum at each timepoint was obtained by removing a 3.00 mL aliquot from the reaction solution and the spectra were collected immediately. The aliquot was readded to the reaction solution, maintaining an effective catalyst concentration throughout the duration of the reaction. The percentage dye concentration remaining was calculated by Equation (1), and Equation (2) gives the percentage degradation:(1)$$\mathrm{percentage}\;\mathrm{dye}\;\mathrm{concentration}=\frac{ [{\mathrm{dye}}_{t}]}{{\mathrm{dye}}_{0}}\times 100$$
(2)$$\mathrm{percentage}\;\mathrm{degradation}=\frac{ [{\mathrm{dye}}_{0}]- [{\mathrm{dye}}_{t}]}{{\mathrm{dye}}_{0}}\times 100$$
where $[{\mathrm{dye}}_{0}]$ and $[{\mathrm{dye}}_{t}]$ are dye concentrations at time 0 and t, respectively, as determined using the aforementioned calibration curves.

The turnover number (TON) of the active catalyst was calculated by the methods employed previously [31,32] and is simply the moles of diazo bonds consumed per mole active site. The turnover frequency (TOF) is given by expressing the TON per unit time (min). To determine the number of active sites on a spherical Ag NP, the number of surface Ag atoms per NP, the number of NPs and finally the number of active sites per reaction must be calculated. The turnover frequency was determined for each dye using 200 mg and 400 mg Ag NP–cotton catalyst [33]. The lattice constant of a face-centered-cubic Ag nanocrystal is 0.409 nm and the volume of the unit cell is (0.409 nm)^3^ = 0.068 nm^3^. Each unit cell of Ag contains four Ag atoms. Each Ag NP has a diameter of 9.7 nm and a radius (r) of 4.85 nm. The volume (V) of a Ag NP sphere is:(3)$$V=\frac{4}{3}{\pi r}^{3}\approx 477.9{\;\mathrm{nm}}^{3}$$

Therefore, there are 28,036 Ag atoms per NP:(4)$$477.9{\;\mathrm{nm}}^{3}\frac{\mathrm{unit}\;\mathrm{cell}}{0.068{\;\mathrm{nm}}^{3}\;}\times \frac{4\;\mathrm{atoms}}{\mathrm{unit}\;\mathrm{cell}}\approx \frac{28,036\;\mathrm{atoms}}{\mathrm{NP}}$$

Given that any one face of the unit cell can face the surface of the NP, then there are two Ag atoms per surface area unit cell, which is (0.409 nm)^2^ = 0.167 nm^2^. The surface area of the NP sphere is given by:(5)$$\mathrm{Surface}\;\mathrm{Area}=4{\pi r}^{2}\approx 296{\;\mathrm{nm}}^{2}$$

The number of Ag atoms (active sites) on the surface of each NP is given by the surface area of the sphere divided by the surface exposed unit cells times the number of Ag atoms per unit cell surface:(6)$$\frac{296\;{\mathrm{nm}}^{2}}{0.167\;{\mathrm{nm}}^{2}}\times 2\;\mathrm{atoms}\approx \frac{3545\;\mathrm{atoms}}{\mathrm{NP}}$$

The number of active sites in the reaction, given the typical 200 mg fabric, was determined. The molecular mass of Ag (107.87 g/mol) with a catalyst load of 13.15 mg/g Ag in cotton fabric, then there were ~2.631 mg Ag/reaction, which is ~24.4 µmol Ag in reaction. From this the number of atoms of Ag can be determined using Avogadro’s number:(7)$$\frac{24.4\times 10^{-6}\;\mathrm{mol}}{\mathrm{reaction}}\times \frac{6.02\times 10^{23}\;\mathrm{atoms}}{\mathrm{mol}}\approx \frac{1.47\times 10^{19}\;\mathrm{atoms}}{\mathrm{reaction}}$$

Since there were 28,036 Ag atoms per NP and there are 1.47 × 10^19^ Ag atoms per reaction, then there are 5.24 × 10^14^ Ag NP per reaction:(8)$$\frac{1.47\times 10^{19}\;\mathrm{atoms}}{\mathrm{reaction}}\times \frac{\mathrm{NP}}{28,036\;\mathrm{atoms}}\approx \frac{5.24\times 10^{14}\;\mathrm{NP}}{\mathrm{reaction}}$$

Given the number of Ag NP per reaction and the number of active sites per NP, it is then straightforward to determine the number of active sites and hence the number of moles of available Ag per reaction:(9)$$\frac{5.24\times 10^{14}\;\mathrm{NP}}{\mathrm{reaction}}\times \frac{3545\;\mathrm{atoms}}{\mathrm{NP}}\times \frac{\mathrm{mol}}{6.02\times 10^{23}\;\mathrm{atoms}}\approx \frac{3.08\times 10^{-6}\;\mathrm{mol}\;\left(\mathrm{active}\right)\;\mathrm{Ag}}{\mathrm{reaction}}$$

When the amount of fabric is doubled (400 mg), all other instances being equal, the available active sites per reaction also double since the number of NPs doubles and the average size of the NP and active sites per NP remain constant.

## 3. Results and Discussion

### 3.1. Characterization of Ag NP–Cotton Catalyst

The UV/vis spectra of pristine white cotton and Ag NPs synthesized within cotton fibers (Figure 1a) as well as the digital microscope images at 50× magnification are shown in Figure 1a–c. When comparing the spectrum of unmodified pristine white cotton with that of the Ag NP cotton catalyst, the presence of a strong absorption band at 420 nm confirmed the formation of Ag NPs [34]. X-ray diffraction (XRD) and X-ray photoelectron spectroscopy (XPS) data confirm the addition of Ag NPs to the cotton fibers and the Ag^0^ valence state in Appendix A, respectively. Additionally, the change in color of the cotton fiber from white to yellow is consistent with the localized surface plasmon resonance of Ag NP formation within the cotton fiber [35]. Internal formation of the Ag NPs within the cotton fiber was confirmed by obtaining the TEM image of the Ag NP–cotton fiber cross-section, Figure 2c, and were observed to be uniformly distributed across the entire fiber diameter and exhibited a spherical morphology. The narrow size distribution for the Ag NPs, 9.7 ± 3.2 nm, Figure 2a,b, can be attributed to the controlled particle growth within the cotton fiber along the cellulose chain [36]. As determined by ICP-MS following acid digestion of the resulting product, the Ag loading was 13.15 mg Ag per gram of cotton.

### 3.2. Catalytic Activity Performance of Ag NP–Cotton

Owing to their remarkable surface area-to-volume ratio, spherical Ag NPs have been shown to have increased chemical reactivity as compared to their bulk metal counterparts because the smaller size results in a greater proportion of metal atoms residing at the surface [37,38]. Here, the catalytic capacity of the synthesized Ag NP–cotton catalyst is studied against the reduction of a mono-azo dye, MO, and two diazo dyes, CR and CSBB, using NaBH_4_ as an electron pair donor in aqueous medium. First, calibration curves for each dye from 50 ppm to 0.1 ppm were obtained by UV/vis spectroscopy to monitor the progress of dye degradation by the Ag NP–cotton catalyst, Figure 3. The UV/vis spectrum for MO is shown in Figure 3a. The strong absorption bands at 464 nm and 496 nm for MO and CR, respectively (Figure 3a,b), are assigned to the n-π* transition of the –N=N– azo-nitrogen lone pair [39,40]. The strong absorption band at 618 nm for CSBB (Figure 3c) corresponds to the π-π* transition of the –N=N– moiety [41,42]. The calibration curves derived from the UV/vis spectra are shown in Figure 3d–f. The decomposition of the dyes by catalytic reduction results in the cleavage of the nitrogen–nitrogen bond and can be monitored by the decrease in the intensity of the respective absorption bands.

The effectiveness of the produced Ag NP–cotton fabric as a catalyst was studied via the catalytic reduction of azo –N=N– bonds in organic dyes. Initially, blank experiments were conducted to confirm that the dyes were not easily reduced by either NaBH_4_ or the Ag NP–cotton catalyst in the absence of the other. The reduction of the dyes by NaBH_4_ in the absence of the Ag NP–cotton catalyst proceeded slowly, with little change to the UV/vis spectra after several hours (Appendix A). Additionally, each dye was found to be stable in solution upon addition of Ag NP–cotton catalyst without NaBH_4_, as determined by UV/vis spectroscopy. There have been several studies investigating the capability of Ag NPs as photocatalysts, capable of reducing azo –N=N– bonds of organic dyes in solution in the absence of an electron donor [43,44]. However, due to the Ag NPs being in the interior of the cotton fiber, light radiation is unable to penetrate through the cotton fiber and initiate the reaction.

Digital images were collected of the dye solutions before and after degradation studies, Appendix A, showing by eye the effectiveness of the reaction to break down the organic dyes in solution. The UV/vis spectra as a function of time for MO is given in Figure 4a using 200 mg Ag NP–cotton catalyst. The absorption band at 464 nm decreased over 90 min with 99.3% dye degradation, Figure 4d. The reaction continued to slowly approach total degradation beyond this time. A proposed catalytic mechanism is presented in Figure 1 [45,46]. Briefly, the electron-rich azo –N=N– double bond coordinates with a surface Ag atom, resulting in inductive electron withdrawal away from the dye nitrogen atom. Subsequently, the nitrogen atom undergoes nucleophilic attack by an electron donor hydride from BH_4_^−^, which is coulombically attracted to the positively charged surface of the Ag NPs, reducing the N=N double bond to a single bond. A second nucleophilic attack by an electron donor hydride from NaBH_4_ results in the cleavage of the N–N single bond. Dissociation from the Ag surface produces the degradation products of sulfanilate and N,N-dimethyl-p-phenylenediamine, and regenerates the Ag NP–cotton catalyst. These degradation products do not interfere with accurate UV/vis measurements since they do not absorb between 300 nm and 750 nm.

The catalytic reduction of the diazo dyes, CR and CSBB, were also monitored by UV/vis spectroscopy at 496 nm and 618 nm, respectively. Under the same conditions, 97.6% and 100% of the dyes were degraded after 145 min and 125 min, respectively (Figure 4b,e and Figure 4c,f, respectively). The longer reaction time for CR and CSBB can be attributed to the presence of two azo moieties in the chemical structure of the molecules. Although the molecular weights of CR and CSBB are greater than that of MO, the presence of two functional groups in close proximity can lead to steric hinderance for reactants in solution. However, the proximity of these functional groups on either side of the central biphenyl synergistically enhances the affinity to the Ag NP surface. As a result, the overall percentage degradation of these dyes was nearly equal to the single azo –N=N– of MO. The TOF was determined for each dye in the dye degradation time plots, and presented in Table 2. For the diazo dyes, the TOF was nearly half of that for MO, at 0.019 min^−1^ and 0.021 min^−1^ compared to 0.041 min^−1^.

The effect of Ag NP–cotton catalyst concentration was investigated by doubling the catalyst loading to 400 mg. The amount of time to complete the reaction significantly decreased for each dye. The degradation of MO reached 96.4% after only 30 min (Figure 5a,d), comparable with the percentage degradation after 90 min using half the catalyst loading. Interestingly, the enhanced reaction rate was most noticeable for the diazo dyes, CR and CSBB (Figure 5b,e and Figure 5c,f, respectively), suggesting their greater affinities assisted in the catalytic reduction by NaBH_4_. However, notably time plots versus percentage degradation for CR and CSBB (Figure 5e,f) exhibit sigmoidal shapes. There is an initial lag in the increase of degradation product. This may be due to the presence of two azo –N=N– moieties where the molar absorptivity of the mono-reduced product is different to that of the diazo product, and therefore the linear region is not reached until only the mono-reduced azo degradation product is present in solution. Unlike 200 mg Ag NP catalyst loading, the TOFs for CR and CSBB, 0.082 min^−1^ and 0.056 min^−1^, respectively, are greater than that of MO (0.046 min^−1^). 

### 3.3. Effect of pH on Catalytic Performance

The effect of solution pH on the degradation of MO, CR, and CSBB by NaBH_4_ using 200 mg Ag NP–cotton catalyst was investigated. Due to the use of NaBH_4_ as a reducing agent, the pH cannot be lowered as it would result in the evolution of hydrogen gas. NaOH was added to the dye solutions to adjust the pH to 9. Slightly decreased reaction times were observed for both MO and CSBB in more alkaline solutions, Figure 6, consistent with previous studies [46,47]. The surface of Ag NPs becomes negatively charged at pH values greater than 8, resulting in electrostatic repulsion between the NP surface and dye. Alternatively, CR showed enhanced reactivity with a decrease in reaction time from 145 to 76 min. Comparing the chemical structures for each of the dyes, MO is minimally affected by the change in pH due to the absence of alkaline sensitive functional groups [46]. Acidic phenylamines are present adjacent to the diazo bond for CR, allowing for electron donating character through resonance, increasing the reactivity for diazo bond degradation. However, CSBB does not have increased reactivity due to the two electron withdrawing sulfate groups on the naphthalene moieties.

### 3.4. Recyclability of Ag NP–Cotton Catalyst

With the Ag NPs produced within the cotton fiber, the Ag NP–cotton catalyst can be easily removed from the reaction solution using tweezers and was then rinsed. Thus, to study the recyclability of the Ag NP catalyst, the same catalyst was used for the catalytic degradation for each dye over 10 cycles. The catalyst maintained very high efficiency (>95% degradation) after 10 cycles (Figure 7). While pristine white cotton fabric permanently changed to the corresponding dye color when immersed in the solution, the Ag NP–cotton catalyst fabric did not change color. Additionally, previous studies have shown that the Ag NPs are well secured within the cotton fiber and are resistant to NP leaching even under aggressive agitation, losing less than 20% of their overall Ag content after 50 laundering cycles. Due to the mild reaction conditions (room temperature and mild mixing by magnetic stirring) it is expected that the catalyst maintains its high silver loading for the lifetime of the fiber [36,48,49], as confirmed by the lack of change in the UV/vis spectrum of the Ag NP–cotton catalyst after 10 cycles (Figure 1).

## 4. Conclusions

This study has shown that internally synthesized Ag NPs within cotton fiber can be used as an effective and efficient catalyst in the reduction of organic azo dyes by NaBH_4_. The reaction was monitored by observing the decrease in the –N=N– absorption bands in the UV/vis spectra. The Ag NP–cotton catalyst was able to degrade highly concentrated solutions of three different azo dyes, a mono-azo and two diazo containing dyes, within a reasonable amount of time. Comparing these results with previous studies, Appendix A, this study demonstrated the capacity of the Ag NP–cotton catalyst to degrade a large volume of organic dye contaminant with relatively low loading. The concentration of the catalyst had a significant effect on the reaction rate, decreasing the overall reaction time and doubling the TOF. Additionally, the recyclability of the Ag NP–cotton catalyst was demonstrated, with little loss in the catalytic efficacy after 10 cycles. The percentage degradation for each dye after 10 cycles was maintained at greater than 95%. This study opens the possibility for a myriad of applications for this Ag NP-embedded cotton catalyst in dye remediation.

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
