# Peer review of "Silver Nanoparticle-Intercalated Cotton Fiber for Catalytic Degradation of Aqueous Organic Dyes for Water Pollution Mitigation"

_nanomaterials, 2022, doi:10.3390/nano12101621_

Round 1

Reviewer 1 Report

Comments are attached

Reviewer 2 Report

The authors have synthesized Ag NPs within cotton fiber. They have used it as an effective and efficient catalyst in the reduction of organic azo dyes by NaBH4. However, the research work is not systematic. And the novelty is not well presented. Therefore, a major modification is required. The specific comments are as follows:

  • The structural characterization of Ag/cotton fiber material is simple, and the particle size, BET, XRD, SEM, etc. should be supplemented with relevant characterizations;
  • The Scheme 1 proposes the possible mechanism of Ag-catalyzed degradation of azo-dyes. Is there any evidence to support it?
  • The current research level in the relevant field should be compared in a table.
  • How is the stability of the catalyst? Can it be recycled and reused?
  • The resolution of the TEM image is too low, it is recommended to report at higher resolution.

Reviewer 3 Report

In this manuscript, authors prepared Ag NPs-incorporated cotton fiber catalyst for effecient degradation of methyl orange (MO), congo red (CR), and Chicago Sky Blue 6B (CSBB) in the presence of sodium borohydride. All contents including preparation, characterization and catalytic property are well organized and presented, I would like to recommend this work for publication in Nanomaterials after minor revision.
1. In Equation (8), where the 3545 atoms/NP came from?
2. In Scheme 1, authors stated that ''H-A represents conjugate acid proton donor.'', but in catalytic experiment, which substance can act a such role, please provide a clear elucidation.
3. In manuscript, ''4'' in NaBH4 should be subscript. 

Reviewer 4 Report

This manuscript reported a photocatalyst silver nanoparticles (Ag NPs) intercalated within cotton fiber for degrading azo dyes (MO, CR, and CSBB). There existed many issues needed to be improved.

  1. More evidences, such as XRD patterns, should be provided to confirm the successful synthesis of the composite.
  2. “The UV/vis spectra as a function of time for MO is given in Figure 4a using 400 mg Ag NP-cotton catalyst” was mentioned in the article. It was inconsistent with the content expressed in Fig 4, it should be “200 mg Ag NP-cotton catalyst”.
  3. “Initially, blank experiments were conducted to confirm the dyes were not easily reduced by either NaBH4 or the Ag NP cotton catalyst in the absence of the other” was mentioned in the manuscript, however, there was no relative data to confirm this result.
  4. The pH value can not only change the surface charge of the catalyst, but also can affect the ionic form of pollutant molecules in the solution. Hence, the influence of pH values on the degradation efficiency should be carried out.
  5. The active species capture experiments should be carried out to better explain the reaction mechanism.
  6. The XRD patterns, SEM, and TEM images before and after the photocatalytic reaction should be provided to further verify the stability of the catalyst.
  7. The XPS spectra of catalyst before and after the photocatalytic reaction should be provided to confirm the valence state of Ag NP and better explain the reaction mechanism.

Round 2

Reviewer 2 Report

This paper is publishable in the present stage

Author Response

We thank the Reviewer for their time and consideration.

Reviewer 4 Report

I recommended that some experimental data should be provided in the revised manuscript. However, the authors cited reported works to reply these questions. Different results would be obtained by different people in different environments. I strongly suggested providing the following experimental data.

  • The XRD patterns of the synthesized samples should be added.
  • The active species capture experiments toward the three pollutants (MO, CR, and CSBB) should be conducted.
  • The XRD patterns and SEM images of samples before and after the photocatalytic reaction should be provided.
  • The XPS spectra of catalysts before and after the photocatalytic reaction should be provided to confirm the valence state of Ag NP and better explain the reaction mechanism.
  • The catalytic mechanism should be comprehensively discussed.

Author Response

I recommended that some experimental data should be provided in the revised manuscript. However, the authors cited reported works to reply these questions. Different results would be obtained by different people in different environments. I strongly suggested providing the following experimental data.

  • The XRD patterns of the synthesized samples should be added.

We have included the XRD and XPS data for the as-synthesized samples used in this study.

  • The active species capture experiments toward the three pollutants (MO, CR, and CSBB) should be conducted.

We agree that this information would be useful and beneficial to overall knowledge, but it is beyond the interested scope of this study. Active species experiments have been conducted by other articles elucidating the active species involvement of methyl orange degradation.[1,2]

  • The XRD patterns and SEM images of samples before and after the photocatalytic reaction should be provided.

We currently have ongoing studies that demonstrate the Ag NPs within the cotton fibers are durable and resist even extensive agitation from 50 commercial laundering cycles in the presence of laundering detergent at 40°C. The conditions describe in this current research are much milder and would resist degradation or aggregation.

  • The XPS spectra of catalysts before and after the photocatalytic reaction should be provided to confirm the valence state of Ag NP and better explain the reaction mechanism.

While the XPS spectra would provide specific information as to the valence state of Ag NP, we find that the UV/vis absorption is a reliable and rapid analysis that provides adequate information as to the oxidation state of the Ag NPs present in the material. Additionally, the cellulose fiber provides stability from oxidation for significant periods of time (several months) and, considering the mild conditions of the reaction, the Ag NPs would remain as Ag0.

  • The catalytic mechanism should be comprehensively discussed.

The catalytic mechanism has been discussed further in the text of the manuscript, per the reviewers suggestions. The difference in chemical structure of the dyes has also been discussed to rationalize the differences in reactivity.

  1. Ilunga, A.K.; Mamba, B.B.; Nkambule, T.T.I. Methyl orange degradation enhanced by hydrogen spillover onto platinum nanocatalyst surface. Applied Organometallic Chemistry 2020, 35, doi:10.1002/aoc.6050.
  2. Li, W.; Li, D.; Lin, Y.; Wang, P.; Chen, W.; Fu, X.; Shao, Y. Evidence for the Active Species Involved in the Photodegradation Process of Methyl Orange on TiO2. The Journal of Physical Chemistry C 2012, 116, 3552-3560, doi:10.1021/jp209661d.

Round 3

Reviewer 4 Report

The relevant data about recommended experiments and measurements still did not provided.

Author Response

We deeply appreciate the expert recommendation of the reviewer. We appreciate the value of these experiments and find value in exploring them further in future work.